# Evidence to Support Inclusion of Pharmacogenetic Biomarkers in Randomised Controlled Trials

**DOI:** 10.3390/jpm9030042

**Published:** 2019-09-01

**Authors:** Danielle Johnson, Dyfrig Hughes, Munir Pirmohamed, Andrea Jorgensen

**Affiliations:** 1Institute of Translational Medicine, Department of Biostatistics, University of Liverpool, Waterhouse Building, 1-5 Brownlow Street, Liverpool L69 3GL, UK; 2Centre for Health Economics and Medicines Evaluation, Bangor University, Ardudwy, Normal Site, Bangor LL57 2PZ, UK; 3MRC Centre for Drug Safety Science and Wolfson Centre for Personalised Medicine, Institute of Translational Medicine, Waterhouse Building, 1-5 Brownlow Street, Liverpool L69 3GL, UK

**Keywords:** pharmacogenetics, biomarker, adverse drug reactions, RCT, evidence

## Abstract

Pharmacogenetics and biomarkers are becoming normalised as important technologies to improve drug efficacy rates, reduce the incidence of adverse drug reactions, and make informed choices for targeted therapies. However, their wider clinical implementation has been limited by a lack of robust evidence. Suitable evidence is required before a biomarker’s clinical use, and also before its use in a clinical trial. We have undertaken a review of five pharmacogenetic biomarker-guided randomised controlled trials (RCTs) and evaluated the evidence used by these trials to justify biomarker inclusion. We assessed and quantified the evidence cited in published rationale papers, or where these were not available, obtained protocols from trial authors. Very different levels of evidence were provided by the trials. We used these observations to write recommendations for future justifications of biomarker use in RCTs and encourage regulatory authorities to write clear guidelines.

## 1. Introduction

The growing field of pharmacogenetics, which studies the effect of genetic biomarkers on the likelihood of treatment response or adverse drug reactions (ADRs) [1], offers an important opportunity to increase the chances of drug benefit and/or reduce the risk of harm [2,3,4,5]. A biomarker is defined as “a characteristic that is objectively measured and evaluated as an indicator of normal biological processes, pathogenic processes, or pharmacologic responses to a therapeutic intervention” [6]. Both germline and somatic genetic biomarkers are being used increasingly to personalise treatments across a wide range of disease areas, including cancer [7,8], thromboembolic disease [9], and autoimmune disease [10], as well as to diagnose disease and provide patient prognosis. 

Many drugs are withdrawn from the market due to lack of efficacy and/or ADRs [11,12,13], and the latter are a major cause of hospital admissions, morbidity, and mortality [14,15]. ADRs are associated with high cost in terms of both time and resources, as well as the negative effects on patient health. There is therefore great potential for genetic biomarker testing to improve the efficacy, safety and cost-effectiveness of medicines. Reviews of economic evaluations of medicines with actionable pharmacogenetic information found the majority of tests to be cost-effective or even cost-saving [16,17]. For example, screening for the *HLA-B*57:01* allele has significantly reduced the incidence of severe ADRs associated with abacavir [18], and has been recommended as a cost-effective intervention [19]. Although it should not be assumed that all pharmacogenetic testing will be cost-effective [20], reductions in the cost of testing and efficiency improvements may see the implementation of more pharmacogenetic tests into clinical practice. 

While the US Food and Drug Administration (FDA) lists over 200 drugs with pharmacogenetic information included in their labels [21], their wider clinical implementation has been limited [22,23,24,25,26]. There are many reasons for this, including the lack of robust evidence of clinical utility [27,28]. Prior to the approval and implementation of biomarker tests in clinical practice, evidence is required of the test’s clinical utility [29,30,31,32] and the gold-standard approach to do this according to guidelines is the randomised controlled trial (RCT) [33,34,35]. A lack of well-designed trials has been cited as one of the main obstacles contributing to the delay in translation of pharmacogenetic discoveries into clinic [28,30,36,37]. Several biomarker-guided trial (‘BM trial’) designs have been proposed for this purpose [38,39,40], and our previously developed online tool, www.bigted.org, provides information about each to guide those designing such a trial [39]. However, before embarking on a BM trial, it is important that robust evidence of the biomarker’s utility and validity is available to justify its inclusion in the trial’s design [41]—without this, there is a risk of wasting money and time on an inappropriate biomarker. Nonetheless, the nature and extent of evidence required, and how it should be compiled, is unclear. More guidance exists on the evidence required for interventions to be included in a trial than for biomarker inclusion, although an integral biomarker assay is just as important a component of the trial [41,42]. 

With this in mind, we undertook a literature review with the aim of reviewing sources of evidence used to justify five previously published pharmacogenetic BM trials. These were chosen to represent different pharmacogenetic biomarker applications. We explored the nature and extent of previous evidence on the association of the included biomarkers with treatment response that had been used to justify their inclusion. We were not concerned with the findings of the trials, instead focusing purely on the evidence cited to justify the inclusion of biomarker(s) within their design. Indeed, we acknowledge that other trials will have been conducted since the publication of the trials included in our review which will have added to the evidence base on the use of the drugs under study. In light of our findings, we also reflected on and provided recommendations on how such evidence should be compiled by those planning future BM trials. 

## 2. Details of Included Trials

To allow us to explore in detail the evidence compiled for each trial, we limited our review to five recently published BM trials. These were chosen carefully to ensure that they were representative of the available trials and spanned a range of different biomarker applications. We felt it important to not only include trials using biomarkers in a way that has been well-characterised (e.g., for targeted therapies), but also those incorporating biomarkers for less well-characterised purposes (e.g., improving medication adherence). The five chosen trials used biomarkers for prevention of ADRs [10], improving efficacy [9], choosing targeted therapies [43], improving medication adherence [44,45], and improving quality of life [46]. Summary details of each trial are provided in Table 1 and full details of data extracted are located in the Appendix A. The first trial (TPMT: AZA Response to Genotyping and Enzyme Testing, TARGET, 2011) explored whether *TPMT* genotyping helped prevent ADRs associated with azathioprine [10,47]. A second trial (European Pharmacogenetics of Anticoagulant Therapy, EU-PACT, 2013) tested whether a genotype-guided approach to calculating therapeutic dose of the anticoagulant, warfarin, led to improved efficacy and reduced the incidence of ADRs [9]. The third trial (SHIVA, 2015) explored the utility of an approach that used genotyping to match patients to molecularly targeted therapies [43]. A fourth trial (Genotype-guided statin therapy, GGST statin trial, 2018) explored whether using genotype testing improved medication adherence and subsequently statin efficacy [44,45,48]. The final trial (NCT02664350) investigated the use of genotyping to reduce pain associated with cancer [46]. 

For each trial, we identified each piece of evidence referenced in the introduction section of a protocol or design paper associated with the trial, and extracted details of the publication year (Figure 1), study design, drug of interest, biomarker used, sample size, country of origin, and the age, sex and ethnicity of participants for each trial. For trials that did not have a published protocol or design paper, we used protocols obtained from contacting the authors (TARGET), or from the results paper Appendix A. Full details of data extracted are found in Table 1. Figures were made using RStudio (version 1.1.453, RStudio Team, Boston, MA, USA) [50], particularly the ‘formattable’ package [51], and LucidChart [52].

## 3. TARGET

TARGET (ISRCTN30748308) began recruitment in 2005 and investigated the use of *TPMT* genotyping to prevent adverse reactions to azathioprine in patients with inflammatory disease [10,53]. The trial randomised inflammatory disease patients (in gastroenterology and rheumatology) 1:1 to genotyping or non-genotyping arms. In the genotyping arm, clinicians were made aware of each patient’s *TPMT* status and the implications of this on dosing prior to commencing azathioprine treatment. Patients in the non-genotyping arm received standard azathioprine dosing. 

TARGET used a biomarker strategy design without biomarker assessment in the control arm [39], Evidence used to justify use of the genotype test spanned the longest time frame of all trials, from 1980 to 2003 (Figure 2). The oldest evidence cited by the trial was a 1980 observational cohort study that proposed a monogenic inheritance pattern for the activity of the TPMT enzyme [54]. Also cited was a 1989 case-control study that compared TPMT enzyme activity in patients who had adverse reactions to thiopurines to a control group that had suffered no reaction [55]. The study showed that patients who had the adverse reaction had extremely low TPMT activity. In total, 11 observational studies were cited, consisting of 9 cohort studies [54,56,57,58,59,60,61,62,63], 1 case control study [55], and 1 study of enzymatic assay use in the UK [64]. A 2001 systematic review of pharmacogenetics in reducing ADRs was cited, although this review was not specific to azathioprine or *TPMT*.

The most recent evidence was an expert opinion by Seidman, 2003 [65]. A 2002 Canadian cost-effectiveness analysis [66], a 2000 case study [67], and a 1997 questionnaire of UK clinicians were also cited [68]. The authors also cited a 2000 guideline from the British Society of Rheumatology, which could not be located online. 

## 4. EU-PACT

The EU-PACT study (NCT01119300) was a large, single-blind, randomised European trial of genotype-guided dosing of warfarin [9,49,69,70,71]. Patients in this trial were randomised 1:1 to genotype-guided or control groups, stratified by centre and treatment indication. Those in the genotype-guided group were genotyped for *CYP2C9* and *VKORC1* and dosed according to an algorithm that included both genetic and clinical factors. The control group received a standard dosing regimen guided by clinical factors only. 

This trial also used a biomarker strategy design without biomarker assessment in the control arm [39]. The published protocol cited mostly observational studies as evidence (Figure 3). These ranged from a 1992 retrospective cohort study [72] to several 2009 studies [73,74,75]. This includes a 2009 genome-wide association study (GWAS) that showed the implications of specific *CYP2C9*, *VKORC1*, and *CYP4F2* genes on warfarin dosing. Also cited were editorials [76,77], cost-effectiveness analyses [78,79], and a literature review of economic evaluations [80]. No previous RCTs were cited. 

## 5. SHIVA

The SHIVA trial (NCT01771458) was a French proof-of-concept histology-agnostic phase II trial using an enrichment design [39] that recruited patients with any metastatic solid cancer to receive treatment with targeted agents [43,81,82]. After analysis of their tumour, patients with mutations that matched an available drug were randomised 1:1 to receive targeted treatment or to physician’s choice treatment.

The total evidence cited in the protocol ranged from 1998 to 2011 (Figure 4). Three RCTs were cited [83,84,85]. Two of these were trials of gefinitib in lung cancer [83,84]. Another RCT cited was an investigation of trastuzumab in HER2+ breast cancer patients, a combination that was investigated in SHIVA [85]. Two observational studies were cited [86,87], along with a contemporaneous editorial commenting on the validity of one of these studies [88].

The paper reporting on the results of this trial included an ‘Evidence before this study’ box [43]. This detailed a literature search performed prior to the start of the trial, which identified several observational cohort studies [87,89,90,91,92] and RCTs [93,94,95].

## 6. GGST Statin Trial

The *SLCO1B1* genotype guided statin therapy (GGST) trial (NCT01894230) investigated the utility of using genotyping to increase adherence to statins and promote lower cholesterol in patients with cardiovascular disease and a history of statin-induced side effects [44,45,48]. Patients were genotyped and then randomised 1:1 to receive genotype information to guide their care, or to usual care alone. The primary outcome in this trial was medication adherence, as assessed by a standard questionnaire. The aim of the trial was to improve adherence by showing patients that treatment includes an assessment of the risks (real and perceived) of statin-induced side-effects [44]. The trial used a biomarker strategy with biomarker assessment in the control arm design [39]. 

This trial cited a large number of references ranging from 2002 to 2015 (Figure 5). Five sets of guidelines from four separate bodies were cited [96,97,98,99,100], alongside an epidemiology report from the American Heart Association [101]. Seven literature reviews were cited [102,103,104,105,106,107,108], alongside two editorials [109,110]. This trial also cited the largest number of observational studies, a total of eleven (consisting of 1 case control study [111], 9 cohort studies [112,113,114,115,116,117,118,119,120], and 1 cohort/meta-analysis study [121]). In contrast to the large amount of observational study evidence, the trial only cited one RCT [122]. Two further references were sub-studies of larger RCTs [123,124]. A 2013 Cochrane review was also cited [125].

The authors cited one systematic review [126] and three meta-analyses [127,128,129]. The systematic review [126] assessed the quality of included studies using ISPOR guidelines [130], and one meta-analysis [129] evaluated quality using the Newcastle-Ottawa scale [131]. The other two meta-analyses were published by the Cholesterol Treatment Trialists’ Collaborators (CTTC) group [127,128], a group established in 1994 to perform meta-analyses of long-term and large-scale trials of lipid intervention therapies [132]. 

The meta-analyses by the CTTC group were both done on the same large data set of *n* = 174,149 participants from 27 RCTs [127,128]. Each RCT had to have a recruitment target of >1000 participants, and have a minimum 2 year treatment duration. The meta-analyses collated individual participant data (IPD). These meta-analyses did not assess the quality of the included studies.

## 7. Precision Medicine Guided Treatment for Cancer Pain

This trial (NCT02664350) used a biomarker strategy design without biomarker assessment in the control arm, and recruited patients with solid tumours and metastases to investigate *CYP2D6*-guided dosing of opioids to manage pain [46]. Patients with pain scores of ≥4 (on a scale of 1–10) were randomised 1:1 to genotype-guided or conventional pain management strategies. This trial did not assign treatments to patients, but provided recommendations to clinicians based on *CYP2D6* genotyping. Patients with poor metabolizer, intermediate metabolizer, or ultra-rapid metabolizer phenotypes were recommended different opioids to those with an extensive (‘normal’) metabolizer phenotype. Those in the control group did not receive *CYP2D6*-guided recommendations. Pain questionnaires were completed at baseline, 2, 4, 6, and 8 weeks. The trial is completed but results have not yet been published.

The authors cited evidence ranging from 1998 to 2017 (Figure 6). The oldest evidence was a 1998 RCT [133], cited alongside 5 other RCTs [134,135,136,137,138]. The newest evidence was 2017 guidelines on adult cancer pain from the National Comprehensive Cancer Network [139]. Interestingly, the trial cited three case studies; one in a patient with the poor metabolizer phenotype [140], and two with patients with the ultra-rapid metabolizer phenotype [141,142]. 

## 8. Discussion

The trials in our review all used different approaches to gathering evidence for justifying biomarker inclusion, and there does not appear to be a standard approach to doing so. Of the trials examined, all cited evidence from within 3 years of their publication (Figure 1). The oldest evidence compared to trial start date was cited by the TARGET trial, which cited work from 25 years prior to its 2005 start date [54].

The evidence types used included systematic reviews/meta-analyses, RCTs, qualitative research, guidelines, recommendations, editorials, and case studies. The traditional ‘evidence pyramid’ is often used to rank evidence types, with meta-analyses and systematic reviews at the top, and case studies and in vitro evidence near the base [143]. However, this has seen some modification in recent years, notably the viewing of evidence through the ‘lens’ of systematic reviews and meta-analyses, ensuring that the quality of included studies is evaluated [144]. In this iteration, a meta-analysis based on weak evidence suffering from bias is not automatically seen as superior evidence to a well-conducted observational study.

To explore the type and extent of evidence compiled to justify including biomarkers in previous BM trials, we have referred to the references in the trial design paper or protocol. This represents a relatively straightforward method of assessing the evidence for a biomarker’s inclusion in a trial, however has some inherent limitations. First, this method will not necessarily capture the entire evidence base upon which inclusion of the biomarker was justified, since the authors may not have provided a complete and accurate snapshot of the evidence they explored and used. Second, journal rules on the amount of references in a paper and word count restrictions could mean that the references included do not represent the totality of evidence used. Similar restrictions on references and word counts may limit the representation of the literature in protocols.

### Recommendations

While the ideal level of evidence is a well-conducted meta-analysis/systematic review of good quality RCTs, including a rigorous assessment of their quality, this is not always available or feasible. In particular, where a biomarker is very new, there may be limited previous evidence to underpin its use. This evidence may take the form of case series or previous case studies. If this is the only evidence available, then this may be the ‘best’ evidence to justify including the biomarker in a trial. It would be important to consider, in such circumstances, whether the proposed RCT would be premature and that the science should first of all be allowed to mature. 

It may be that different standards of evidence may be necessary for different biomarker types [25,145]. For example, evidence standards could be based on risk, with biomarkers for lower risk applications requiring less evidence and regulatory oversight than those for high risk applications [145]. Recommendations could also be based on the disease being treated, similar to how orphan drugs for rare diseases are given accelerated approvals [146,147]. Biomarkers used for more serious indications could be allowed to proceed to trial with less or lower quality evidence than biomarkers for less serious conditions. Novelty of the biomarker will also influence the extent of evidence available—for example a biomarker first utilised in 1980 is likely to have accumulated much more evidence than one first described in 2015. 

Further, some conditions have existing diagnostic or treatment guidance algorithms that do not use biomarkers but have good clinical utility. In these scenarios, adding a biomarker to the algorithm might provide a low value of information compared to a biomarker used in a condition where a good clinical algorithm is not available. Therefore, authors might consider prioritizing the development of biomarkers for conditions that do not have sufficient clinical prediction methods for diagnosis or guiding treatment. 

It is also important to ensure that genetic biomarkers are not subject to higher evidentiary requirements than other types of biomarkers. This ‘genetic exceptionalism’ and the higher burden of evidence for genetic tests has been shown to be a barrier to implementation [4,25,30,148]. Finally, biomarkers that are integral to a trial’s conduct require more evidence than biomarkers used on an exploratory basis [41]. 

With these factors in mind, our recommendations for all biomarker-guided trials consist of two essentials (Figure 7).
-Systematic review before embarking on a trial

We would recommend an initial systematic review is undertaken prior to the start of any trial. The Lancet journal now requires all research papers to include a ‘Research in Context’ panel that shows the evidence available prior to the study, and how the authors searched for this information [149]. This is an important step that should be considered by all journals and particularly any source funding a clinical trial. The search should be supplemented with evidence from other sources such as clinical guidelines and pilot data. 

Regardless of the type of evidence identified in the systematic review, we recommend that the quality of that evidence is assessed when justifying including the biomarker, and we suggest that design-specific tools are used for this purpose (e.g., the Cochrane Collaboration’s Risk of Bias tool for RCTs) [150]. Several study type-specific methods for doing this are available [131,150,151,152,153,154] and have been reviewed by Zeng, et al. (2015) [155]. We additionally recommend the quality of pharmacogenetic studies is assessed using the guidelines proposed by Jorgensen and Williamson (2008) [156].

When synthesising evidence already existing from previous studies, it is also important to consider the age and ethnicities of the populations of the previous studies compared to the proposed trial’s population to ensure that the evidence is relevant. Many studies (94% in one review [157]) imply generalisability of results without acknowledging the differential effects of race and ethnicity. Differences in cancer incidence, stage at discovery [158], and mortality [159] have been found to be functions of race or ethnicity and it is imperative that trialists consider the ethnicity of the proposed trial population and to keep this in mind when evaluating the evidence relating to biomarker validity. Notably, a 2016 review found that 81% of participants in genome-wide association studies were white [160], and several studies have shown that non-white people are less likely to be clinical trial participants [157,161,162] and are less likely to access genetic testing services [163]. It is important, therefore, in a drive to reduce such inequalities, that the clinical utility of ethno-specific biomarkers are tested in trials recruiting participants from relevant ethnic backgrounds. Similar considerations should be given to other factors known to contribute to health inequalities, including age, gender, and socio-economic position. These factors are summarized by the PROGRESS-Plus acronym recommended by the Cochrane Public Health Group [164].

Further, if the systematic review reveals a sufficient number of previous RCTs or observational studies, authors should consider conducting a meta-analysis to assess the current evidence quantitatively. This would help ascertain whether there was sufficient uncertainty surrounding the current evidence to justify the planned RCT. An example of where this could have been implemented is in the fifth trial we examined [46]. Authors can also utilise funnel plots to examine any potential bias in the publication of included studies [165], and explore any heterogeneity between studies.
-Guidelines are required

Given the lack of standardisation across BM trials in terms of how inclusion of biomarkers are justified, we recommend that guidelines are developed to aid researchers in compiling and presenting evidence to justify their inclusions. This will not only ensure that sufficient evidence exists prior to embarking on a BM trial, thus avoiding waste of resources, but will also serve as a useful guide to those planning a BM trial and provide transparency in the trial report. 

The Clinical Pharmacogenetics Implementation Consortium (CPIC) provides guidelines for the implementation of pharmacogenetics [166]. The guidelines provide a grading of the level of evidence given in support of the biomarker’s implementation (‘high’, ‘moderate’ or ‘weak’) [167]. The CPIC levels are based on PharmGKB criteria (Figure 8), where the evidence for a gene-drug association is rated on a six-point scale between 1A (guidelines endorsed by a medical society or major health system) to 4 (in vitro, case study, or nonsignificant study evidence) [29]. This scale is based on ‘clinical annotations’ obtained from PubMed, produced by combining and summarising associations from several publications. These clinical annotations are then given a ‘level of evidence’ score based on replication of the association, *p*-value, and odds ratio. The score is determined by PharmGKB curators [29].

Whilst these guidelines are for implementation of biomarkers into clinical practice in a patient who has a known genotype, a similar approach could be developed for justification of use in a RCT. We located one paper that discussed the incorporation of biomarkers into early phase clinical trials [41], but we recommend that this needs to contribute to the formation of formal guidelines for BM trials similar to CPIC guidelines for biomarker implementation.

Finally, the conclusions and recommendations above are based on the assumption that a BM trial is indeed required. It is possible that when compiling the evidence to justify inclusion of a biomarker in a trial that it is so overwhelmingly in favour of the biomarker’s clinical utility that it may be unethical to restrict its use to a randomised trial. This loss of clinical equipoise is something important to consider and indeed clinical implementation may be recommended and accepted without the need for a BM trial in such cases. 

## Figures and Tables

**Figure 1 jpm-09-00042-f001:**
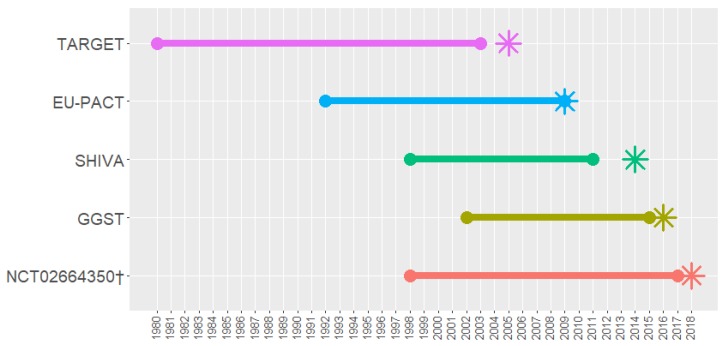
Timings of publications cited by each trial. Star icons indicate the date of publication of the paper or protocol references were extracted from. †results not yet published.

**Figure 2 jpm-09-00042-f002:**
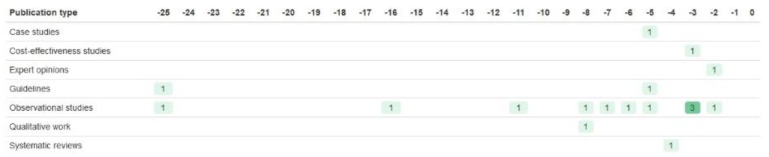
Evidence cited by the TARGET trial to justify inclusion of the *TPMT* biomarker, relative to the publication of the 2005 protocol [47].

**Figure 3 jpm-09-00042-f003:**
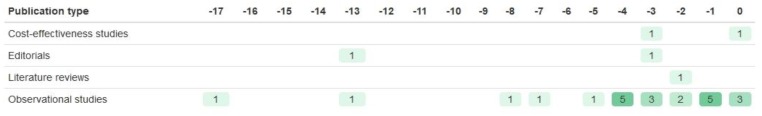
Evidence cited by the EU-PACT trial to justify inclusion of the *CYP2C9* and *VKORC1* biomarkers, relative to the publication of the 2009 published protocol [49].

**Figure 4 jpm-09-00042-f004:**
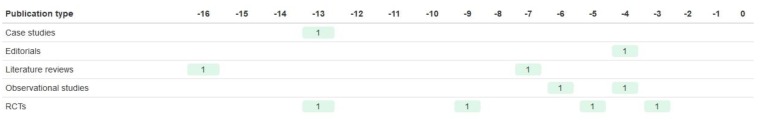
Evidence cited by the SHIVA trial to justify inclusion of the biomarkers, relative to the publication of the 2014 protocol (included in Supplementary of a 2015 paper [43]). RCT = randomised controlled trial.

**Figure 5 jpm-09-00042-f005:**
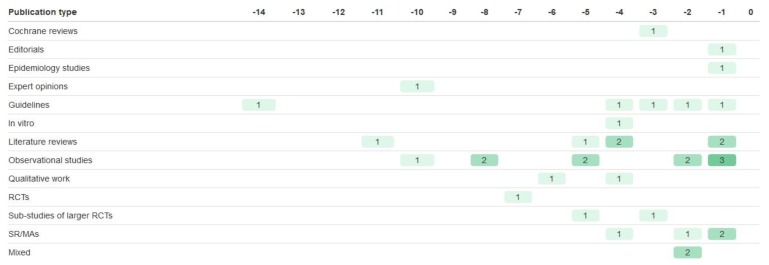
Evidence cited by the GGST statin trial to justify inclusion of the SLCO1B1 biomarker, relative to the publication of the 2016 rationale and design paper [44]. ‘Mixed’ refers to papers that used a mixture of two or more of the other publication types. RCT = randomised controlled trial SR/MAs = systematic reviews/meta-analyses.

**Figure 6 jpm-09-00042-f006:**
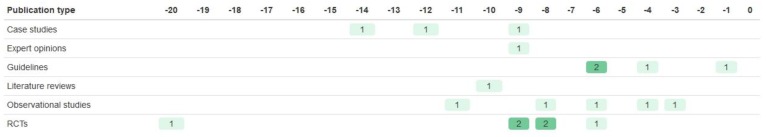
Evidence cited by the NCT02664350 trial to justify inclusion of the CYP2D6 biomarker, relative to the publication of the 2018 design and rationale paper [46].

**Figure 7 jpm-09-00042-f007:**
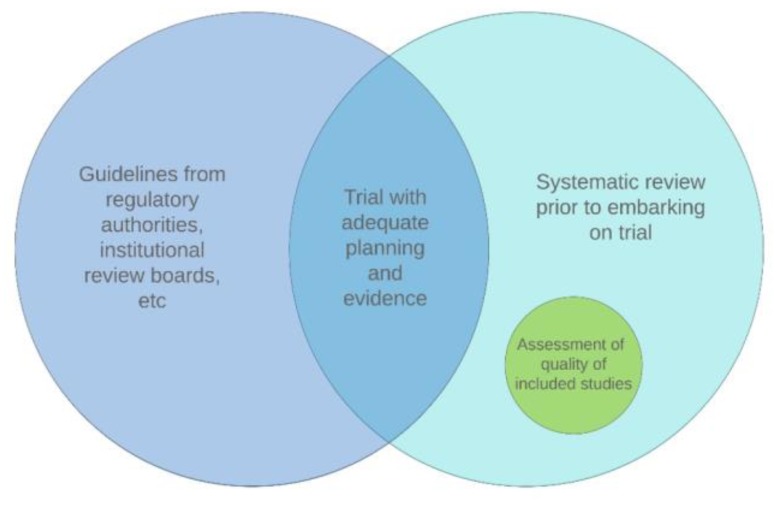
Our recommendations for evidence gathering prior to the start of a biomarker-guided trial, based on the findings of this review.

**Figure 8 jpm-09-00042-f008:**
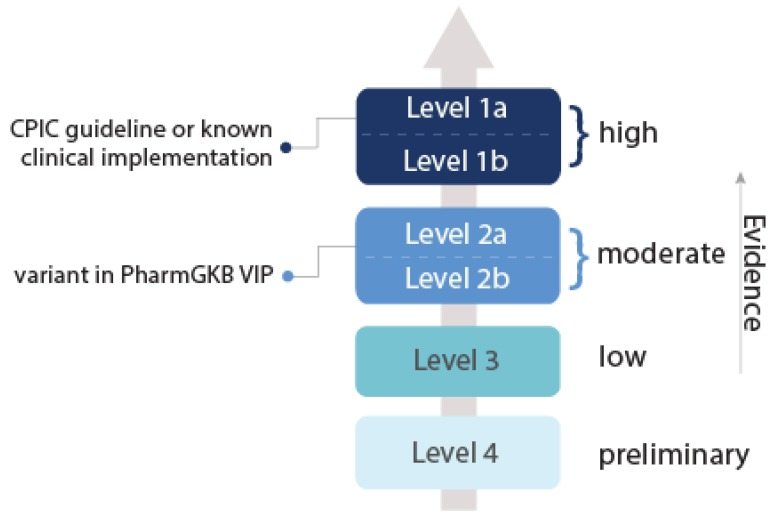
Guidelines of the Clinical Pharmacogenetics Implementation Consortium (CPIC) for the grading of biomarker evidence, based on the PharmGKB evidence criteria [29,168].

**Table 1 jpm-09-00042-t001:** Details of selected trials. Start year denotes year the first patient was recruited. BM trial (biomarker-guided trial) design is the design as selected by using the BiGTeD online resource [39].

Registration Number	Trial Name	Start Year	Year of Results Publication	Paper References Taken from	BM Trial Design	Biomarker	Drug of Interest	Sample Size (n Randomised)	Age of Participants	Sex of Participants	Ethnicity of Participants	Study Location
ISRCTN30748308	TARGET (protocol) [10,47]	2005	2011	2005 protocol obtained from authors	Biomarker strategy design (without biomarker assessment in control arm)	*TPMT*	Azathioprine	333	Mean 43.2 (control)	50.6%/49.4% F/M (control)	92.2% white, 4.8% South Asian, 0.6% Black, 2.4% mixed/other (control)	UK
									Mean 41.0 (genotyped)	50.3%/49.7% F/M (genotyped)	89.8% white, 7.2% South Asian, 3.0% Black, 0% mixed/other (genotyped)	
NCT01119300	EU-PACT [49]	2011	2013	2009 paper 10.2217/pgs.09.125	Biomarker strategy design (without biomarker assessment in control arm)	*CYP2C9*2*	Warfarin	455	Mean 66.9 (control)	42.1%/57.9% F/M (control)	98.7% white, 0.9% Black, 0.4% Asian (control)	UK, Sweden
						*CYP2C9*3*			Mean 67.8 (genotyped)	35.8%/64.2% F/M (genotyped)	98.2% white, 1.3% Black, 0.4% Asian (genotyped)	
						*VKORC1*						
NCT01771458	SHIVA [43] (protocol)	2012	2015	2014 protocol obtained from authors	Enrichment design	Hormone receptors pathway	Targeted chemotherapy agent, based on genotyping	195	Median 63 (control)	72%/28% F/M (control)	Not reported	France
						*PI3K/AKT/mTOR* pathway			Median 61 (genotyped)	61%/39% F/M (genotyped)		
						*RAF/MEK* pathway						
NCT01894230	GGST statin trial [44]	2013	2018	2016 paper 10.2217/pgs-2016-0065	Biomarker strategy design (with biomarker assessment in control arm)	*SLCO1B1*5*	Any statin	159	Mean 62.5 (control)	65.8%/34.2% F/M (control)	80.3% white, 14.5% Black, 5.3% other (control)	USA
									Mean 62.7 (genotyped)	49.4%/50.6% F/M (genotyped)	79.5% white, 16.9% Black, 3.6% other (genotyped)	
NCT02664350	n/a [46]	2016	Results not yet published	2018 paper 10.1016/j.cct.2018.03.001	Biomarker strategy design (without biomarker assessment in control arm)	*CYP2D6*	Opioids	200 (forecast)	Not available	Not available	Not available	USA

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
