# Peer review of "Evidence to Support Inclusion of Pharmacogenetic Biomarkers in Randomised Controlled Trials"

_jpm, 2019, doi:10.3390/jpm9030042_

Round 1

Reviewer 1 Report

This is a well written review of several different RCTs examining pharmacogenetic biomarkers. The following are minor considerations for the authors and editor. 

1) Include more explicit detail as to why each article was chosen so as to explain to the reader that they were not just cherry-picked to make an explicit point. 

2) Consider explaining the difference between non-genotyped vs genotyped and control vs. genotyped. 

3) There are two parts on page 4 where there is an "Error! Reference source not found."

4) Consider using the year on the X-axis for figures 2-6 consistent with figure 1 rather than relative to the year published. 

5) On page 7 in the first paragraph of section 7 consider calling patients "normal" rather than extensive metabolizers, or putting "normal" in parenthesis. 

6) On page 10 in the first paragraph, consider expanding to include more detail on which ethnicities/age groups PGx studies may be biased towards. 

Author Response

Dear Reviewer 1,

We thank you for your useful feedback on our work. Below we have addressed each comment in turn.

1) Include more explicit detail as to why each article was chosen so as to explain to the reader that they were not just cherry-picked to make an explicit point.

We thank the reviewer for this useful feedback. Indeed the articles were carefully chosen and not cherry-picked, and as such we have added some additional wording to the first paragraph in section 2 to ensure that our reasoning behind choosing the trials is more transparent.

2) Consider explaining the difference between non-genotyped vs genotyped and control vs. genotyped.

Thank you, we have amended these terms in Table 1 to remove any ambiguity. All ‘non-genotyped’ are now specified as ‘control’ groups instead.

3) There are two parts on page 4 where there is an "Error! Reference source not found."

We thank the reviewer for pointing out these errors with the referencing, and have corrected them.

4) Consider using the year on the X-axis for figures 2-6 consistent with figure 1 rather than relative to the year published.

We thank the reviewer for this suggestion, however we feel that the timing of the various pieces of supporting evidence relative to the trial is more important and informative than their respective dates of publication in the context of figures 2-6. Because of this, and given that the publication year of the pieces of evidence are available elsewhere in the manuscript, we have not changed the x-axis for these figures.

5) On page 7 in the first paragraph of section 7 consider calling patients "normal" rather than extensive metabolizers, or putting "normal" in parenthesis.

We thank the reviewer for this advice, and agree that identifying this subset of patients as ‘normal’ is useful and consistent with terminology commonly used in the literature. As such, we have amended the first paragraph of section 7 to:  “Patients with poor metabolizer, intermediate metabolizer, or ultra-rapid metabolizer phenotypes were recommended different opioids to those with an extensive (‘normal’) metabolizer phenotype.”

6) On page 10 in the first paragraph, consider expanding to include more detail on which ethnicities/age groups PGx studies may be biased towards.

We thank the reviewer for this suggestion and have made some changes to the first paragraph on page 10 to provide more context on wider biases in research and why it is important for those planning a biomarker-guided trial to consider them.

Additionally -

Figure 1 has also been amended slightly by replacing the asterisks next to the wording ‘NCT02664350’ on the y-axis with a dagger (†), to avoid confusion in the caption of referring to the ‘star icons’ on the plot.

Reviewer 2 Report

Discuss ethno-specific genetic biomarkers and their utility to reduce current disparities of care. Current guidelines do not often provide enough granulation as to the relevance of such population-dependent variants in order to make actionable recommendations for underrepresented groups.

Since the utility and clinical value of a pharmacogenetic biomarker decreases when current predictive ability of existing tests is high, authors should consider a prioritization approach to be followed for biomarkers in conditions without the clinical ability (methods) to adequately predict response.

Minor concerns:
Figures 2-6: could these figures be merged to Figure 1?

Author Response

Dear Reviewer 2,

We thank you for your useful feedback on our work, and we have addressed each comment in turn below.

Discuss ethno-specific genetic biomarkers and their utility to reduce current disparities of care. Current guidelines do not often provide enough granulation as to the relevance of such population-dependent variants in order to make actionable recommendations for underrepresented groups.

We thank the reviewer for this feedback, and we have added a paragraph on page 10 on the wider context of bias in research, and how these are important for those planning a trial to consider. We have focussed on race/ethnicity bias, but also acknowledge other factors affecting health inequalities.

Since the utility and clinical value of a pharmacogenetic biomarker decreases when current predictive ability of existing tests is high, authors should consider a prioritization approach to be followed for biomarkers in conditions without the clinical ability (methods) to adequately predict response.

Thank you for this interesting point. We have added a paragraph on page 9 noting this need for prioritisation.

Minor concerns:

Figures 2-6: could these figures be merged to Figure 1?

We thank the reviewer for this suggestion, however we feel that each figure should be made available directly after the text that refers to it. We also believe that the timing of the pieces of evidence relative to their trial is more important than their respective dates of publication in the context of figures 2-6, which is why they are needed in addition to Figure 1.

Additionally - 

Figure 1 has also been amended slightly by replacing the asterisks next to the wording ‘NCT02664350’ on the y-axis with a dagger (†), to avoid confusion in the caption of referring to the ‘star icons’ on the plot.